# Antimicrobial Resistance Profiles and Macrolide Resistance Mechanisms of *Campylobacter coli* Isolated from Pigs and Chickens

**DOI:** 10.3390/microorganisms9051077

**Published:** 2021-05-17

**Authors:** Ji-Hyun Choi, Dong Chan Moon, Abraham Fikru Mechesso, Hee Young Kang, Su-Jeong Kim, Hyun-Ju Song, Soon-Seek Yoon, Suk-Kyung Lim

**Affiliations:** Bacterial Disease Division, Animal and Plant Quarantine Agency, 177 Hyeksin 8-ro, Gimcheon-si 39660, Gyeongsangbuk-do, Korea; wlgus01@korea.kr (J.-H.C.); ansehdcks@korea.kr (D.C.M.); abrahamf@korea.kr (A.F.M.); kanghy7734@korea.kr (H.Y.K.); kimsujeong27@gmail.com (S.-J.K.); shj0211@korea.kr (H.-J.S.); yoonss24@korea.kr (S.-S.Y.)

**Keywords:** *C*. *coli*, food animals, macrolide, mutation, resistance, virulence

## Abstract

We identified 1218 *Campylobacter* *coli* isolates from fecal and carcass samples of pigs (*n* = 643) and chickens (*n* = 575) between 2010 and 2018. About 99% of the isolates were resistant to at least one antimicrobial agent. The isolates exhibited high resistance rates (>75%) to ciprofloxacin, nalidixic acid, and tetracycline. Azithromycin and erythromycin resistance rates were the highest in isolates from pigs (39.7% and 39.2%, respectively) compared to those of chickens (15.8% and 16.3%, respectively). Additionally, a low-to-moderate proportion of the isolates were resistant to florfenicol, gentamicin, clindamycin, and telithromycin. Multidrug resistance (MDR) was found in 83.1% of the isolates, and profiles of MDR usually included ciprofloxacin, nalidixic acid, and tetracycline. We found point mutation (A2075G) in domain V of the *23S rRNA* gene in the majority of erythromycin-resistant isolates. Multilocus sequence typing of 137 erythromycin-resistant *C*. *coli* isolates revealed 37 previously reported sequence types (STs) and 8 novel STs. M192I, A103VI, and G74A substitutions were frequently noted in the ribosomal proteins L4 or L22. Further, we identified a considerable proportion (>90%) of erythromycin-resistant isolates carrying virulence factor genes: *flaA*, *cadF*, *ceuE*, and *VirB*. The prudent use of antimicrobials and regular microbiological investigation in food animals will be vital in limiting the public health hazards of *C*. *coli* in Korea.

## 1. Introduction

*Campylobacter* species are commensal bacteria that reside within the gastrointestinal tract of many wild and domestic animals. They are among the most important foodborne pathogens that cause human gastroenteritis worldwide [1]. *Campylobacter* was associated with 236 foodborne outbreaks between 2010 and 2017 in the United States [2]. In European countries, human cases of campylobacteriosis have exceeded those caused by classic enteric bacteria such as *Salmonella* or *Escherichia coli*, with about 214,000 confirmed cases reported in 2016 [3]. Additionally, data from low and middle-income countries indicated that the rate of *Campylobacter* infection has increased over the past decade [4,5,6].

Although most cases of *Campylobacter* enteritis are self-limiting, severe or prolonged cases of enteritis, septicemia, and other extraintestinal infections may require antibiotic treatment [4]. Fluoroquinolones are commonly used to treat human campylobacteriosis. In addition, macrolides such as azithromycin and erythromycin are drugs of choice for infections caused by fluoroquinolone-resistant *Campylobacter* strains. However, the increase in the consumption of antimicrobials in food animals has contributed to the emergence of antimicrobial-resistant *Campylobacter* strains. Indeed, the observation of *Campylobacter* strains that are resistant to critically important antimicrobials in food animals has raised concerns that treatment of human infections will be compromised [1,4].

Macrolides inhibit bacterial RNA-dependent protein synthesis by targeting the 50S ribosomal subunit. The binding of macrolides leads to conformational changes in the ribosome and subsequent termination of the elongation of the peptide chain [7,8]. Base substitutions in multiple alleles of the *23S rRNA* gene are the most common mutations conveying macrolide resistance in *Campylobacter* species [9]. Mutations at positions 2074 and 2075 in the peptidyl transferase region in domain V of the *23S rRNA* target gene are associated with high-level macrolide resistance (MIC > 128 µg/mL) in *C. coli* [8,9]. Mutations have also been identified in the ribosomal proteins L4 and L22, both of which form portions of the polypeptide exit tunnel within the bacterial 70S ribosome and have been described in *C*. *coli* [8,10]. In addition, macrolide resistance has been associated with the chromosomally-encoded multidrug-resistant efflux system and ribosomal methylation encoded by the erythromycin ribosome methylase B-*erm*(*B*) gene [9,10]. Notably, the CmeABC multidrug efflux pump is reported to work in synergy with specific mutations, even in the absence of any other factor affecting resistance [11].

The incidence of *Campylobacter* infection is increasing worldwide [6]. *C*. *coli* is considered the second most common *Campylobacter* species responsible for human infections, next to *C*. *jejuni*, and continues to present a significant threat to food safety and public health. In the past decade, macrolide-resistant *C. coli* isolates of animal origin were reported in many countries [12,13,14,15]. Despite frequent reports of macrolide resistance in *C. coli* isolated from food animals and humans in South Korea (Korea) [16,17,18,19,20], only a few studies were performed to determine the resistance mechanisms [21,22]. The studies were conducted in a relatively small number of isolates collected from some specific parts of the country before 2016. Considering the global public health risk posed by macrolide-resistant *C. coli* in food animals and the increase in the total consumption of macrolides in food animals in Korea [23], continuous investigation of the resistance profiles and the mechanisms of macrolide resistance in *C. coli* isolated from food animals is vital to safeguard public health. Therefore, we performed extensive evaluations of the antimicrobial resistance profiles, the mechanism(s) of macrolide resistance, and virulence factor genes in *C. coli* isolated from fecal and carcass samples of chickens and pigs in Korea from 2010 to 2018.

## 2. Materials and Methods

### 2.1. Collection and Identification of C. coli

Altogether 1218 *C. coli* isolates (643 pig and 575 chicken isolates) were obtained from 16 laboratories/centers participating in the Korean Veterinary Antimicrobial Resistance Monitoring System from 2010 to 2018 (Appendix A). *C*. *coli* was isolated from the feces and carcasses of pigs, and chickens. One to five isolates were collected from each farm. The isolation of *C*. *coli* was performed using Bolton broth (Thermo Scientific, Basingstoke, UK) and *Campylobacter* blood-free selective agar (Thermo Scientific, Basingstoke, UK), as previously described [19]. Isolates were then confirmed using the polymerase chain reaction as described by Denis et al. [24]. However, we do not have information about the number of slaughterhouses, animals, and samples considered for this study.

### 2.2. Antimicrobial Susceptibility Test

Antimicrobial susceptibility was determined via the broth microdilution method according to the Clinical and Laboratory Standards Institute [25] guideline, using commercially available antibiotic-containing CAMPY plates (Sensititre, Trek Diagnostics, Cleveland, OH, USA). *C. jejuni* ATCC 33560 was used as a reference strain. The following nine antimicrobials were tested: azithromycin (0.015–64 µg/mL), ciprofloxacin (0.015–64 µg/mL), clindamycin (0.03–16 µg/mL), erythromycin (0.03–64 µg/mL), florfenicol (0.03–64 µg/mL), gentamicin (0.12–32 µg/mL), nalidixic acid (4–64 µg/mL), telithromycin (0.015–8 µg/mL), and tetracycline (0.06–64 µg/mL). Antimicrobial resistance breakpoints were determined based on the National Antimicrobial Resistance Monitoring System [26]. Multi-drug resistance (MDR) was defined as resistance to three or more antimicrobial subclasses. One erythromycin-resistant isolate per farm was considered for further characterization.

### 2.3. Analysis of Macrolide Resistance Mechanisms

A total of 137 isolates (87 from pigs and 50 from chickens) were selected from different farms for analysis of macrolide resistance mechanisms and subsequent characterization. A PCR assay was used to investigate the presence of *erm(B)* gene as previously described [27]. Mutations in the genes encoding domain V of the *23S rRNA* and ribosomal proteins L4 and L22 were determined as previously described [10,11]. Briefly, PCR reactions were performed in a final volume of 20 µL containing genomic DNA, PCR mix, and each of the forward and reverse primers (Solgent, Daejeon, Korea). After an initial denaturation of 5 min at 95 °C, amplification was performed over 30 cycles each consisting of 95 °C for 1 min, annealing temperature for 1 min, and 72 °C for 1 min with a final extension of 7 min at 72 °C. PCR products were purified (Solgent, Daejeon, Korea) and sequenced using an ABI prism 3100 analyzer (Genotech, Daejeon, Korea). Analysis of the nucleotide sequence and comparison with *C. coli* JV20 genome (GeneBank accession number NZ_AEER01000024) were performed using the BLAST program (http://www.ncbi.nlm.nih.gov/BLAST (accessed on 12 Auguts 2020)) and ExPASY proteomics tools (http://www.expasy.ch/tools/#similarity (accessed on 12 Auguts 2020)).

### 2.4. Detection of Virulence Factor Genes

We analyzed virulence factor genes linked with *Campylobacter* motility (*flaA*), adhesion and invasion (*cadF*, *dnaJ*, *pldA*, *racR*, *virB*, *ceuE*, and *ciaB*), cytotoxic production (*cdtA*, *cdtB*, and *cdtC*), and Guillain-Barré syndrome (*wlaN*) using PCR, as previously described [28,29,30].

### 2.5. Multilocus Sequence Typing (MLST) and eBURST Analysis

MLST was performed according to Dingle et al. [31]. Specific primers (Genotech, Daejeon, Korea) were used to amplify and sequence the following 7 housekeeping genes: aspA, glnA, gltA, glyA, pgm, tkt, and uncA. PCR products were purified (Solgent, Daejeon, Korea) and sequenced using an ABI prism 3100 analyzer (Genotech, Daejeon, Korea). Allele profiles and sequence types (ST)s were designated using the MLST website for Campylobacter (https://pubmlst.org/organisms/campylobacter-jejunicoli) (accessed on 14 April 2021). Each sequence is assigned with an allele number, and the combination of alleles yields an ST. In addition, the relatedness of the sequence types was determined using goeBURST software (http://goeBURST.phyloviz.net (accessed on 19 April 2021))

### 2.6. Statistical Analysis

Antimicrobial resistance rates and Pearson correlation were analyzed using Excel (Microsoft Excel, 2016, Microsoft Corporation, Redmond, WA, USA). *P* values less than 0.05 were considered statistically significant.

## 3. Results

### 3.1. Antimicrobial Resistance

The majority of *C*. *coli* isolates (>75%) recovered from pigs and chickens exhibited high resistance rates to ciprofloxacin, nalidixic acid, and tetracycline (Table 1). Indeed, the highest resistance rates were observed in chicken isolates compared to that of pigs. More than one-third of the pig isolates were resistant to azithromycin, clindamycin, erythromycin, and telithromycin. Additionally, relatively small percentages (<18%) of chicken isolates exhibited resistance to these antimicrobials. Florfenicol resistance was noted only in 8.4% and 1.7% of pig and chicken isolates, respectively.

### 3.2. Antimicrobial Resistance Trends

The resistance rates of most of the tested antimicrobials in pig and chicken isolates remained stable (Table 1). The gentamicin resistance rate in chicken isolates relatively peaked in 2016–2018. Despite fluctuations, we noted a trend of decreasing resistance (*P* < 0.05) to telithromycin in pig isolates. Additionally, the florfenicol resistance rate remained very low throughout the study period, especially in chicken isolates.

### 3.3. Antimicrobial Resistance Patterns

In this study, 98.9% (1204/1218) of the isolates were resistant to one or more of the tested antimicrobials (Table 2). MDR was noted in the majority of pig (83.8%) and chicken (82.3%) isolates (Table 1 and Table 2). We identified 57 and 27 different resistance patterns in chicken and pig isolates, respectively (Appendix A). Resistance to seven of the tested antimicrobials, except to florfenicol and gentamicin, was the major MDR pattern in pig isolates, whereas resistance to ciprofloxacin, nalidixic acid, and tetracycline was the predominant MDR pattern in chicken isolates. Notably, five isolates from pigs and three isolates from chickens exhibited resistance to all of the tested antimicrobials.

### 3.4. Detection of Mutation and ermB Gene

Analysis of the *23S rRNA* gene among erythromycin-resistant isolates (*n* = 137) demonstrated an A2075G mutation in 83 (95.4%) and 46 (92%) pig and chicken isolates, respectively (Table 3). Erythromycin-resistant isolates from pigs (A2075G and C2097T, *n* = 2) and chickens (A2075G and T2114C, *n* = 4; A2074M and A2075Y, *n* = 1) exhibited double mutations. However, no mutation was found in the *23S rRNA* gene in three of the pig and chicken isolates, each.

We identified various types of mutations in the ribosomal proteins L4 and L22 in erythromycin-resistant isolates from chickens and pigs (Table 3). Pig isolates exhibited seven types of amino acid substitutions in the ribosomal proteins L4 (M192I, *n* = 31; V176I, *n* = 16, T177S, *n* = 16; V184I, *n* = 16; V121A, *n* = 14; P28S, *n* = 7; and A140T, *n* = 1) and one in ribosomal protein L22 (A103V, *n* = 11). In addition, chicken isolates presented four types of amino acid substitutions in the ribosomal proteins L4 (M192I, *n* = 4; V184I, *n* = 2; P28S, *n* = 1; and V121A, *n* = 2) and six in ribosomal protein L22 (G74A, *n* = 5; V65I, *n* = 4; T109A, *n* = 4; Q24R, *n* = 2; T109S, *n* = 1; and V65M, *n* = 1). Notably, we identified mutations in both the *23S rRNA* gene and L4 and/or L22 ribosomal protein (s) in 43 pig and nine chicken isolates. However, none of the investigated isolates from chickens and pigs harbored the *erm(B)* gene.

### 3.5. Virulence Factor Genes

The erythromycin-resistant isolates were investigated for the presence of various virulence factor genes that are associated with *Campylobacter* motility, adhesion, and invasion into human intestinal cells, and cytotoxin production. More than 85% of the isolates carried at least three virulence factor genes (Table 4). The majority (>93%) of isolates from pigs carried the *cadF*, *ceuE*, *flaA*, and *virB* genes. Similarly, the *cadF*, *ceuE*, and *flaA* genes were detected in at least 90% of the isolates from chicken. However, the *virB* gene was detected in only 4.1% of chicken isolates. None of the isolates carried genes associated with cytotoxin production, expression of Guillain–Barré syndrome, and most of the invasion-associated virulence factors.

### 3.6. MLST and eBURST Analysis

We identified 37 and 16 different STs from the 87 pig and 50 chicken isolates, respectively (Table 3). The dominant STs in pigs were ST854 (*n* = 12), ST1016 (*n* = 8), ST2715 (*n* = 7), and ST829 (*n* = 6), whereas ST860 (*n* = 16) and ST9867 (*n* = 9) were frequent in chickens. Thirteen STs in pigs (four ST1556, four ST11054, three ST828, three ST890, three ST1055, three ST11050, two ST1096, two ST1131, two ST1142, two ST1450, two ST2733, two ST9575, and two ST4172) and five STs in chickens (four ST6148, four ST9201, three ST5675, three ST11051, and two ST1016) were each represented by fewer than five isolates. In addition, 20 STs in pigs and 9 STs in chickens were represented by only a single isolate each. Among the identified STs, six from pigs (ST11049, ST11050, 11054, ST11056, ST11061, and ST11062) and two from chickens (ST11051 and 11052) were reported for the first time. Moreover, eight of the STs (ST860, ST828, ST829, ST854, ST1016, ST1096, ST1556, and ST2715) were identified in both chicken and pigs. Further, the goeBURST algorithm revealed 38 STs with five or more allele matches, whereas seven STs are singletons and are unrelated to any other within the single locus variant collection (Figure 1).

## 4. Discussion

Our data demonstrated that a considerable proportion of chicken and pig *C*. *coli* isolates were resistant to some of the clinically important antimicrobials. We found diverse STs, mutations, and virulence factor genes in the majority of erythromycin-resistant isolates.

The macrolide resistance rate was high in pig isolates compared to that of chicken isolates. *C*. *coli* isolated from various food animals and carcasses in Korea presented variable resistance rates to azithromycin (30–43%) and erythromycin (6–30%) [16,17,18,19,20]. In this study, the azithromycin and erythromycin resistance rates in pig and chicken isolates were higher than those reported in Europe [32,33,34]. In contrast, our findings in chicken isolates were lower than those described in Africa [5,9], China [35,36], and some European countries [15,37]. The use of macrolides for the prevention and control of various diseases in food animals, particularly pigs, in Korea could be associated with the emergence of erythromycin and azithromycin-resistant *C*. *coli*. Indeed, about 68% of the total macrolide sold for livestock in Korea is used in pig husbandry [23]. These observations are concerning because macrolides, especially erythromycin and azithromycin, are the drug of choice for the treatment of human *Campylobacter* infections [1,4].

In agreement with previous studies in Korea [16,17,18,19] and China [35,36], *C*. *coli* isolated from pigs and chickens exhibited very high resistance rates to ciprofloxacin, nalidixic acid, and tetracycline. However, our findings were much higher than previous reports in Africa [5,38], Europe [32,34], and North America [39,40]. The frequent use of fluoroquinolones and tetracyclines in food animals can select resistant strains that could be readily transferred to humans through the food chain [41].

The gentamicin resistance rate in *Campylobacter* species has been reported to be low [34]. In this study, the gentamicin resistance rates in chicken and pig isolates were higher than previous reports in Africa [12,42], the EU [32], and North America [40], but it was low compared to those reported in China [35,36]. Gentamicin is normally considered for serious bacteremia and other systemic infections due to *Campylobacter* [8]. Thus, the observation of resistance to this antibiotic in considerable proportions of isolates from food animals has a potential public health implication.

Globally, the incidences of resistance to several key antibiotics useful in the treatment of *Campylobacter* disease are increasing and multiple resistance patterns to several classes of antibiotics are emerging [8]. High levels of multidrug resistance among *C*. *coli* isolates have been observed within the food chain [43]. In our study, about 83% of the isolates were resistant to at least three antimicrobial classes. Previous studies in Thailand [44], Poland [43], and France [45] revealed that 99%, 95%, and 54% of *C*. *coli* isolates, respectively, from various food animals and carcasses were resistant to multiple antimicrobials. In China, 42% to 98% of *C*. *coli* isolated from retail chicken exhibited MDR [14,36]. Concordant with previous reports in Guatemala [46], Europe [33], and the United States [47], profiles of MDR usually included ciprofloxacin, nalidixic acid, tetracycline, and to some extent erythromycin. Alfredson et al. [8] revealed that trends in antimicrobial resistance have shown a clear association between the use of antibiotics in food animals and resistant isolates of *Campylobacter* in humans. The increasing incidence of resistance to several key antibiotics in *C*. *coli* presented a public health threat [8].

The study showed that antimicrobial resistance rates of *C*. *coli* isolated from pigs and chickens in Korea differed from those described previously from various geographical regions. However, comparing and contrasting data between studies is often difficult due, for example, to different origins, duration of studies, number of isolates studied, and laboratory analysis. The differences in antimicrobial use in livestock husbandry among countries could also contribute to the variation in the prevalence of antimicrobial resistance.

Base substitutions at positions 2074 and 2075 of the adenine residues in the *23S rRNA* gene in *Campylobacter* spp. are the most common mutations associated with erythromycin resistance [48]. In this study, we identified A2075G mutation in the majority of erythromycin-resistant isolates. A2075G mutation in all three copies of the *23S rRNA* gene is associated with high-level macrolide resistance [48], as it has been indicated in previous studies in many countries, including Korea [9,21,22,49]. In addition, this mutation has been shown to provide stability to *Campylobacter* in culture and maintain their ability to colonize their host [50]. We also detected double mutations in two pig (A2075G and C2097T) and five chicken (A2075G and T2114C, A2074M, and A2075Y) isolates. Despite previous reports on the detection of double mutation in erythromycin-resistant *C*. *jejuni* and *C*. *coli* isolated from humans [51,52], these types of double mutations have been rarely associated with erythromycin-resistant *Campylobacter* species isolated from food animals.

Amino acid substitutions in the ribosomal proteins L4 and L22 are linked with a low level of macrolide resistance in *Campylobacter* species. Amino acids at positions 63–74 are a part of the most important target region in ribosomal protein L4 [11]. However, in this study, no variation was found in this region. Most of the changes were concentrated in the region at amino acid 121-192, except in eight strains that harbored the P-28→S replacement. Among the seven types (V121A, V176A, T177S, V184I, M192I, A140T, and P28S) of substitutions identified in the ribosomal protein L4, M192I was the most frequent change observed in *C. coli* isolates recovered from food animals, especially pigs. Previous studies [21,53] have also identified V121A, T177S, and M192I substitutions in ribosomal protein L4 in erythromycin-resistant isolates from various sources. In this study, mutation in ribosomal protein L22 was not common compared to ribosomal protein L4. We noted seven types of mutations in ribosomal protein L22: A103V, Q24R, V65I, G74A, T109A, V65M, and T109S. A103V was the predominant type of substitution found only in pig isolates. Consistent with this study, V65I, A74G, and A103V substitutions were identified in erythromycin-susceptible and -resistant isolates in Korea and other countries [10,21,53]. Substitutions in the ribosomal proteins L4 and L22 are known to confer low-level resistance to macrolides [10,11,50]. Further, we observed the coexistence of mutation in the *23S rRNA* gene and amino acid substitutions in L4 and/or L22, although the significance of the coexistence is unknown.

Three erythromycin-resistant isolates from chickens identified in this study did not harbor any mutation. Furthermore, none of the isolates carried the *erm(B)* gene. Although we did not investigate other resistance mechanisms, the presence of efflux pumps could be linked with erythromycin resistance [49]. Wei and Knag [21] identified erythromycin-resistant *Campylobacter* strains that did not exhibit any of the currently identified resistance mechanisms, indicating the presence of unidentified mechanisms. Therefore, further studies are needed to elucidate mechanisms underlying the development of resistance.

MLST followed by eBURST clustering is useful for assessing major changes of the lineages among isolates and is suitable for periodic typing and global epidemiology [14]. We found 45 various STs; 37 were from pigs and 16 were from chickens. Indeed, 8 of the STs (ST860, ST828, ST829, ST854, ST1016, ST1096, ST1556, and ST2715) were identified in both chicken and pigs. Previous studies in Korea [14,22,54] and other countries [55,56,57,58,59] have reported diverse STs in *C*. *coli* isolated from food animals and humans, and ST827, ST828, ST829, and ST855 were the predominant STs. The most frequent STs identified in this study (ST854, ST1016, ST2715, ST829, ST860, and ST9867) differed from those described in previous reports [55,56,57,58,59], except for ST829 *C*. *coli*, which was frequently detected in pigs in Korea [22]. Genetic diversity in the *Campylobacter* population might emerge through mutation and recombination events [59]. Chicken isolates shared the same STs with that of pigs, indicating the dissemination of identical clones in the poultry and pig industry. In addition, the identification of new STs in pigs (ST11049, ST11050, ST11054, ST11056, ST11061, and ST11062) and chickens (ST11051 and ST11052) might suggest the emergence of new clones in the poultry and pig industry. Among the identified *C*. *coli* STs, ST828, ST829, ST860, ST872, ST1055, ST1058, and ST1446 were frequently reported in patients with campylobacteriosis [55,56]. Thus, food animals may serve as an important reservoir and source of human infection. Furthermore, the presence of closely related STs may indicate the evolutionary relationship between isolates [54,59].

The expression of genes involved in *Campylobacter* motility, adhesion and invasion into intestinal epithelial cells, as well as toxin production, is vital for the establishment of infection in humans. Motility of the bacterium is fundamental for adhesion into intestinal epithelial cells in the early stage of pathogenesis. We identified the flagellin-coding *flaA* gene, which is primarily responsible for bacterial motility [60], in at least 90% of the erythromycin-resistant isolates recovered from pigs and chickens. These findings are in agreement with previous reports in Poland [61] and Vietnam [62]. We also detected the *cadF* gene, which encodes for a fibronectin-binding outer membrane protein, in almost all of the erythromycin-resistant isolates. The *cadF* gene is responsible for bacterial adhesion and influencing microfilament organization in host cells [28]. Many virulence factors have been associated with the invasion of *Campylobacter* into intestinal epithelial cells, including the *pldA*, *virB*, *iam*, *ceuE,* and *ciaB* [63]. The *ceuE* gene was identified in almost all of the erythromycin-resistant isolates. Although the exact mechanism remains obscure, it is one of the most important genes encoding for *Campylobacter* invasion [30]. Further, we noted the *virB* gene in 93.1% of pig and 4% of chicken isolates. The *virB* gene encodes a putative type IV secretion system involved in adhesion and invasion of *Campylobacter* to the intestinal epithelial cells [64].

In conclusion, the present study demonstrated long-term trends that *C*. *coli* isolated from food animals exhibits resistance to multiple clinically important antimicrobials. We identified erythromycin-resistant *C*. *coli* isolates with diverse STs and various mutations in the *23S rRNA* gene and ribosomal proteins L4 and L22. Our observations highlight the need for proper food safety practices to prevent the spread of antimicrobial-resistant and virulent strains of *Campylobacter* spp. Additionally, the prudent use of antimicrobials in food animals and constant monitoring of resistance among *Campylobacter* isolates in food animals and animal products are urgently needed.

## Figures and Tables

**Figure 1 microorganisms-09-01077-f001:**
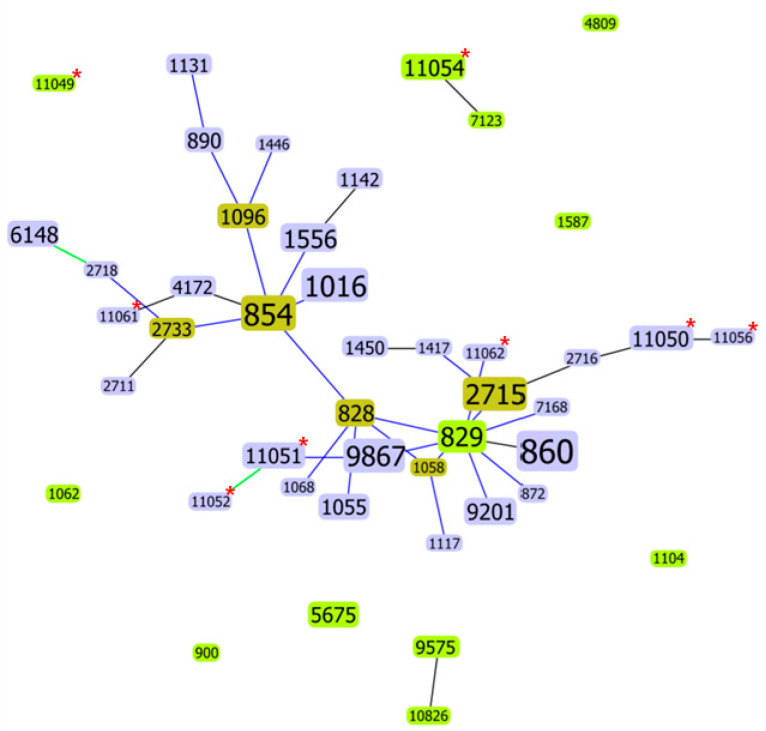
goeBURST analysis conducted at a single locus variant level, with double locus variants added to show further relatedness. Light green ST nodes, group founder; dark green ST nodes, subgroup founder; light blue ST nodes, common node; and red asterisks, novel STs. Colored links: black, link drawn without recourse to tiebreak rules; blue, link drawn using tiebreak rule 1; and green, link drawn using tiebreak rule 2.

**Table 1 microorganisms-09-01077-t001:** Antimicrobial resistance profiles of *C. coli* isolated from pigs and chickens from 2010 to 2018 in Korea.

Antimicrobials	% (No.) of Resistant Isolates										
	Pigs					Chickens					Total
	2010–2012	2013–2015	2016–2018	Subtotal	*p*-Value	2010–2012	2013–2015	2016–2018	Subtotal	*p*-Value	(*n* = 1218)
	(*n* = 268)	(*n* = 263)	(*n* = 112)	(*n* = 643)		(*n* = 196)	(*n* = 103)	(*n* = 276)	(*n* = 575)		
Azithromycin	35.8 (96)	42.2 (111)	42.9 (48)	39.7 (255)	0.2763	16.3 (32)	12.6 (13)	16.7 (46)	15.8 (91)	0.9436	28.4 (346)
Ciprofloxacin	91.4 (245)	86.3 (227)	88.4 (99)	88.8 (571)	0.6020	99 (194)	98.1 (101)	99.3 (274)	98.8 (568)	0.8456	93.5 (1139)
Clindamycin	40.7 (109)	46 (121)	42.9 (48)	43.2 (278)	0.7289	16.8 (33)	11.7 (12)	17.8 (49)	17.2 (99)	0.9023	31 (377)
Erythromycin	35.1 (94)	42.2 (111)	42 (47)	39.2 (252)	0.3491	15.8 (31)	11.7 (12)	16.3 (45)	16.3 (94)	0.9368	28.4 (346)
Florfenicol	5.2 (14)	10.3 (27)	11.6 (13)	8.4 (54)	0.2102	0.5 (1)	1.9 (2)	2.5 (7)	1.7 (10)	0.1445	5.3 (64)
Gentamicin	11.6 (31)	14.4 (38)	12.5 (14)	12.9 (83)	0.7961	13.3 (26)	19.4 (20)	28.6 (79)	21.7 (125)	0.0741	17.1 (208)
Nalidixic acid	91.4 (245)	85.9 (226)	83.9 (94)	87.9 (565)	0.1675	98.5 (193)	97.1 (100)	99.3 (274)	98.6 (567)	0.7661	93 (1132)
Telithromycin	46.6 (125)	40.7 (107)	33.9 (38)	42 (270)	0.0260	12.2 (24)	12.6 (13)	14.9 (41)	13.6 (78)	0.2457	28.6 (348)
Tetracycline	77.6 (208)	82.5 (217)	70.5 (79)	78.4 (504)	0.5995	78.6 (154)	89.3 (92)	81.2 (224)	80.9 (465)	0.8503	79.6 (969)
MDR	84.3 (226)	85.8 (226)	76.9 (86)	83.8 (539)		77.7 (151)	85.4 (88)	84.8 (234)	82.3 (473)		83.1 (1012)

MDR, multidrug resistance.

**Table 2 microorganisms-09-01077-t002:** Frequent resistance patterns in *C. coli* isolated from pigs (*n* = 643) and chickens (*n* = 575) from 2010 to 2018 in Korea.

Pigs		
Number of Antimicrobials	% (No.) of Isolates	Most Frequent Resistance Pattern
0	1.6 (10)	-
1	4.2 (27)	TET (*n* = 23)
2	10.4 (67)	CIP NAL (*n* = 56)
3	33.7 (217)	CIP NAL TET (*n* = 187)
4	8.7 (56)	CIP GEN NAL TET (*n* = 14)
5	6.2 (40)	CIP CLI FFC NAL TET (*n* = 13)
6	6.1 (39)	AZM CIP CLI ERY NAL TEL (*n* = 16)
7	21.5 (138)	AZM CIP CLI ERY NAL TEL TET (*n* = 131)
8	6.8 (44)	AZM CIP CLI ERY GEN NAL TEL TET (*n* = 37)
9	0.8 (5)	AZM CIP CLI ERY FFC GEN NAL TEL TET (*n* = 5)
MDR	83.8 (539)	
**Chickens**		
**Number of** **Antimicrobials**	**% (No.) of Isolates**	**Most Frequent Resistance Pattern**
0	0.7 (4)	-
1	0.3 (2)	CIP (*n* = 2)
2	16.7 (96)	CIP NAL (*n* = 94)
3	50.4 (290)	CIP NAL TET (*n* = 269)
4	16 (92)	CIP GEN NAL TET (*n* = 79)
5	0.5 (3)	CIP FFC GEN NAL TET (*n* = 1)
		AZM CIP GEN NAL TET (*n* = 1)
		AZM CIP CLI ERY NAL (*n* = 1)
6	4 (23)	AZM CIP CLI ERY NAL TET (*n* = 12)
7	8 (46)	AZM CIP CLI ERY NAL TEL TET (*n* = 42)
8	2.8 (16)	AZM CIP CLI ERY GEN NAL TEL TET (*n* = 15)
9	0.5 (3)	AZM CIP CLI ERY FFC GEN NAL TEL TET (*n* = 3)
MDR	82.3 (473)	

Abbreviations: AZM, azithromycin; CIP, ciprofloxacin; clindamycin, CLI; ERY, erythromycin; FFC, florfenicol; GEN, gentamicin; NAL, nalidixic acid; TEL, telithromycin; TET, tetracycline; MDR, multidrug resistance.

**Table 3 microorganisms-09-01077-t003:** Mutations in erythromycin-resistant *C. coli* isolated from pigs and chickens from 2010 to 2018 in Korea.

Source	ERY ^a^ MIC Range (µg/mL)	Nucleotide/Amino Acid Substitution				Sequence Types
		23s rRNA ^b^	L4^c^	L22 ^c^	No. of Isolates	
Pigs (*n* = 87)	≥64	A2075G	WT	WT	38	7 ST854, 7 ST1016, 2 ST2715, 2 ST2733, 2 ST11050, 2 ST1556, 2 ST4172, 2 ST828, and each of ST860, ST900, ST1062, ST1068, ST1104, ST1142, ST1446, ST11056, ST11049, ST11061, ST2716, and ST890
	>64	A2075G	WT	A103V	1	ST854
	>64	A2075G	WT	no band	1	ST10826
	≥64	A2075G	M192I	WT	13	4 ST829, 2 ST1055, and each of ST828, ST872, ST1142, ST2715, ST7168, and ST11054, ST11062
	64	A2075G	M192I	A103V	3	ST1131, ST1450, and ST2715
	>64	A2075G	P28S	WT	6	2 ST854, 2 ST9575, and each of ST1556 and ST11050
	>64	A2075G	V121A	WT	2	ST1096 and ST11054
	>64	A2075G	V121A, V176I, T177S, V184I, M192I	WT	4	ST1016, ST1096, ST7123, and ST11054
	>64	A2075G	V121A, V176I, T177S, V184I, M192I	A103V	4	ST1417, ST1450, ST1556, and ST11054
	>64	A2075G	V176I, T177S	WT	2	ST1058 and ST1117
	>64	A2075G	V176I, T177S, V184I	WT	1	ST2715
	>64	A2075G	V176I, T177S, V184I, M192I	WT	2	ST854 and ST2715
	>64	A2075G	V176I, T177S, V184I, M192I	A103V	1	ST1131
	>64	A2075G	V184I, M192I	WT	2	ST829 and ST2715
	>64	A2075G	no band	no band	1	ST829
	>64	A2075Y	WT	WT	1	ST4809
	>64	A2075G, C2097T	V121A, V176I, T177S, V184I, M192I	A103V	2	ST890
	>64	WT	V121A, A140T	WT	1	ST2718
	>64	WT	V121A	WT	1	ST1055
	>64	WT	P28S	WT	1	ST854
Chickens (*n* = 50)	≥64	A2075G	WT	WT	32	10 ST860, 7 ST9867, 4 ST9201, 2 ST5675, 2 ST6148, 2 ST11051, 2ST1016, and each of ST828, ST1587, and ST1556
	>64	A2075G	WT	Q24R, V65I, G74A, T109A	2	ST11052 and ST860
	64	A2075G	WT	V65I, G74A, T109A	1	ST5675
	>64	A2075G	M192I	G74A, T109A	1	ST2711
	>64	A2075G	P28S	WT	1	ST854
	>64	A2075G	V184I	WT	1	ST1096
	64	A2075G	V121A, M192I	V65M	1	ST11051
	>64	A2075G	V121A, M192I	V65I, G74A, T109S	1	ST6148
	64	A2075G	V184I, M192I	WT	1	ST2715
	>64	A2075G	no band	WT	1	ST860
	≥32	A2075G, T2114C	WT	WT	4	ST860
	>64	A2074M, A2075Y	WT	WT	1	ST6148
	≥32	WT	WT	WT	3	2 ST9867 and an ST829

^a^ Abbreviations: ERY, erythromycin; WT, wild type; ST, ^b^ position according to *Escherichia Coli* numbering. ^c^ Position of amino acids changes. DNA sequences of *rplD* and *rplV* genes coding L4 and L22 ribosomal proteins, respectively, were compared with the sequence in the *C*. *coli* JV20 genome.

**Table 4 microorganisms-09-01077-t004:** Prevalence of virulence marker genes in erythromycin-resistant *C. coli* isolated from pigs and chickens from 2010 to 2018 in Korea.

Source	No. of Isolates	Distribution (%) of Virulence Factor Genes															
		*flaA*	*cadF*	*dnaJ*	*pldA*	*racR*	*virB*	*ceuE*	*ciaB*	*cdtA*	*cdtB*	*cdtC*	*wlaN*	1	2	3	4
Pigs	87	93.1	97.7	0	0	0	93.1	98.8	0	0	0	0	0	1.1	4.6	89.6	4.6
Chickens	50	90	100	0	0	0	4	100	0	0	0	0	0	0	10	86	4

## Data Availability

The data that support the findings of this study are available from the corresponding author upon reasonable request.

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
