# Peer review of "Antimicrobial Resistance Profiles and Macrolide Resistance Mechanisms of Campylobacter coli Isolated from Pigs and Chickens"

_microorganisms, 2021, doi:10.3390/microorganisms9051077_

Round 1

Reviewer 1 Report

The authors responded to my previous comments and. After careful reading of the revised manuscript, I have few minor comments and suggestions.

  1. Authors performed MLST typing of macrolide resistant isolates (n=137).
  • Authors should mention the number of analyzed isolates for MLST , e.g.:
  • L19: multilocus sequence typing of 137 macrolide resistant C. coli isolates revealed…
  • L162: Analysis of the 23S rRNA gene among macrolide resistant isolates ( n=137) demonstrated…
  1. They found 45 various STs, which clearly excluded clonality of this set of C. coli.
  • This important finding should be mentioned in related parts of manuscript, e.g., abstract, discussion.
  1. Other minor comments:
  • L33: number “214, 000” change to “214000”
  • L39, L332: “Campylobacteriosis” should be without capital letter
  • Table 4, first column: check the font discrepancies (bold)
  • L264: “Guatemalla” check the size of letters

Author Response

Comments to the authorThe authors responded to my previous comments and. After careful reading of the revised manuscript, I have few minor comments and suggestions.

Authors performed MLST typing of macrolide-resistant isolates (n=137).

Point 1. Authors should mention the number of analyzed isolates for MLST, e.g.:

L19: Multilocus sequence typing of 137 macrolide-resistant C. coli isolates revealed¡¦

Response: Thank you for the suggestion and it is revised as suggested (Line 19)

L162: Analysis of the 23S rRNA gene among macrolide-resistant isolates (n=137) demonstrated¡¦

Response: Thank you for the suggestion and it is revised as suggested (Line 164)

Point 2. They found 45 various STs, which clearly excluded clonality of this set of C. coli.

This important finding should be mentioned in related parts of manuscript, e.g., abstract, discussion.

Response: Thank you for the suggestion. Information on the sequence types ae included in lines 20 (abstract), 207(results), and 323-324 (discussion). In addition, a brief description of the total isolates considered for MLST and subsequent molecular characterization is included in the methods (lines 100 and 101).

Other minor comments:

Point 3. L33: number "214, 000" change to "214000"

Response: corrected accordingly

Point 4. L39, L332: "Campylobacteriosis" should be without capital letter

Response: corrected accordingly

Point 4. Table 4, first column: check the font discrepancies (bold)

L264: "Guatemalla" check the size of letters

Response: corrected accordingly

Reviewer 2 Report

The manuscript has been improved to a great extend. All the points were addressed appropriately. I think this new version is ready for publication.

Author Response

Response: Thank you for taking the time to review our manuscript.

This manuscript is a resubmission of an earlier submission. The following is a list of the peer review reports and author responses from that submission.

Round 1

Reviewer 1 Report

The current study represents the antimicrobial resistance pattern of a large number of isolated Campylobacter coli in South Korea. The authors have also provided useful information regarding the possible mechanisms of different antibiotic resistance. Although the study is interesting; however, my main criticism applies to studying the C. coli isolates and not the C. coli strains. In this case, we have no idea what percentage of outcome refers to different strains and what percentage refers to repeating the same repeated strains (it can be misleading).  It is a pity that the authors did not discriminate the strains before studying these large number of isolates. Maybe they have an explanation for that. The material and method are poorly described, it needs to be improved too. Please see below the technical comments:

-line 64, in this section, the information regarding the Campylobacter isolation should be complete, whether by citation or explanation (the media, antibiotic used in media, atmosphere, etc.).

-lines 75-76, please provide the proper references for the following technique you used: broth dilution method using 75 Sensitire panel KRNV4F (EUST, TREK Diagnostic Systems, West Sussex, UK).

-line 80, please mention the name of the company for antibiotics.

-line 82, the reference provided here (26), is not related to antimicrobial resistance breakpoints determination by NARMS.

-line 87, the reference provided here (27), is not related to any gene resistance detection by PCR. Please recheck the citations.

-line 85, I think it is more appreciated if the authors explain the method they have used in their study. Microorganisms journal policy requires as many details as possible for material and methods to make it simple for paper readers. Here in this section, one citation was irrelevant, and the other one had cited another citation itself for that particular protocol.

-line 128, these eight resistant isolates to all antimicrobial agents should carry some acquired gene resistance. It would be very interesting to go further and sequence some genome for further studies. Of course, after discrimination of strains using a molecular approach in case (just suggestion this part).

- I think table 2 is confusing. These tables can change into a graphical figure. The authors can show these data in a donut chart, which makes it more fascinating.

- line 189, can we link the high rate of mutations in oriental countries to the differences between East and West? Do we have any information on this issue in the literature?

-line 228, did the double mutation caused the resistance, or the resistant gene without any mutation could do it? How can the authors say that the double mutation caused the resistance? What about the effect of these mutations on the MIC value? Please clarify it.

Reviewer 2 Report

In the manuscript “Antimicrobial resistance profiles and macrolide resistance mechanisms of Campylobacter coli isolated from pigs and chickens in South Korea authors analyzed an antibiotic resistance in an extensive set of veterinary Campylobacter coli isolates. They observed a high resistance of C. coli to several antimicrobial agents, including several MDR-profiles. Moreover, authors performed a sequence analysis of macrolide-resistant isolates and identified several mutations in genes related to the macrolide resistance.

This is a very interesting topic since Campylobacter infections are global human and veterinary health problem. The strength of this manuscript is in a higher number of isolates colected from pigs and chickens (more than 1000 isolates) and complementary microbial and molecular analysis of antibiotic resistances; on the other hand, the epidemiological data and characterization of isolates are insufficient.

 The manuscript is well-written and easy to understand. I have a few comments, which could improve the manuscript.

Major comments:

  1. Authors analyzed the antibiotic resistance in set of 1218 isolates, which originated from 16 various places, but there is lack of additional epidemiological data. Several various antibiograms in the set of isolates were identified, but clonal character of isolates was not excluded. At the same time, the clonality is well documented among MDR-enterobacteria, such as ESBL- coli isolates. Authors should perform a typing of isolates (e.g. MLST, PFGE, rep-PCR) and exclude their clonal character, at least for dominant C. coli variants; for example, for large group of MDR groups with same patterns (see Table 2) and dominant groups with identical macrolide-resistance patterns (see Table 3).
  2. In addition, typing of isolates allows to compare them with other studies and interpretation of their relevance for human infections. The relevance of these isolates with human coli infections should be discussed in a greater detail.

Minor comments:

  1. The Table 2 is confusing and should be improved. Since there is no difference in ATB resistance during the years, these data should be removed to supplementary material. The most frequent pattern should be presented in a Figure rather than in table format.